# Spiking Neural Networks for Multimodal Neuroimaging: A Comprehensive Review of Current Trends and the NeuCube Brain-Inspired Architecture

**DOI:** 10.3390/bioengineering12060628

**Published:** 2025-06-09

**Authors:** Omar Garcia-Palencia, Justin Fernandez, Vickie Shim, Nicola Kirilov Kasabov, Alan Wang

**Affiliations:** 1Auckland Bioengineering Institute, The University of Auckland, Auckland 1010, New Zealand; ogar209@aucklandui.ac.nz (O.G.-P.); j.fernadez@auckland.ac.nz (J.F.); v.shim@auckland.ac.nz (V.S.); nkasabov@aut.ac.nz (N.K.K.); 2Centre for Brain Research, The University of Auckland, Auckland 1010, New Zealand; 3Mātai Medical Research Institute, Gisborne 4010, New Zealand; 4Knowledge Engineering and Discovery Research Innovation, School of Engineering, Computer and Mathematical Sciences, Auckland University of Technology, Auckland 1010, New Zealand; 5Knowledge Engineering Consulting Ltd., Auckland 1071, New Zealand; 6Institute for Information and Communication Technologies, Bulgarian Academy of Sciences, 1113 Sofia, Bulgaria; 7Faculty of Medical and Health Sciences, The University of Auckland, Auckland 1010, New Zealand; 8Centre for Co-Created Ageing Research, The University of Auckland, Auckland 1010, New Zealand

**Keywords:** spiking neural networks, neuroimaging, brain-inspired, NeuCube, neuromorphic computing

## Abstract

Artificial intelligence (AI) is revolutionising neuroimaging by enabling automated analysis, predictive analytics, and the discovery of biomarkers for neurological disorders. However, traditional artificial neural networks (ANNs) face challenges in processing spatiotemporal neuroimaging data due to their limited temporal memory and high computational demands. Spiking neural networks (SNNs), inspired by the brain’s biological processes, offer a promising alternative. SNNs use discrete spikes for event-driven communication, making them energy-efficient and well suited for the real-time processing of dynamic brain data. Among SNN architectures, NeuCube stands out as a powerful framework for analysing spatiotemporal neuroimaging data. It employs a 3D brain-like structure to model neural activity, enabling personalised modelling, disease classification, and biomarker discovery. This paper explores the advantages of SNNs and NeuCube for multimodal neuroimaging analysis, including their ability to handle complex spatiotemporal patterns, adapt to evolving data, and provide interpretable insights. We discuss applications in disease diagnosis, brain–computer interfaces, and predictive modelling, as well as challenges such as training complexity, data encoding, and hardware limitations. Finally, we highlight future directions, including hybrid ANN-SNN models, neuromorphic hardware, and personalised medicine. Our contributions in this work are as follows: (i) we give a comprehensive review of an SNN applied to neuroimaging analysis; (ii) we present current software and hardware platforms, which have been studied in neuroscience; (iii) we provide a detailed comparison of performance and timing of SNN software simulators with a curated ADNI and other datasets; (iv) we provide a roadmap to select a hardware/software platform based on specific cases; and (v) finally, we highlight a project where NeuCube has been successfully used in neuroscience. The paper concludes with discussions of challenges and future perspectives.

## 1. Introduction

This section provides an in-depth exploration of how an ANN is reshaping neuroimaging analysis, addressing both its transformative potential and inherent challenges. These insights lay the foundation for introducing brain-inspired SNNs as a promising alternative, which will be further elaborated in subsequent sections.

### 1.1. Overview of AI in Neuroimaging

AI is revolutionising the field of neuroimaging analysis, offering powerful tools to extract meaningful information from complex brain scans and advance our understanding of the brain. The most significantly affected areas are as follows:

Automated Image Analysis: AI algorithms enhance neuroimaging analysis by automating tasks, such as segmentation [1], feature extraction [2], and anomaly detection [3]. Segmentation allows AI to precisely differentiate tissue types or identify specific brain structures, minimising the need for manual labelling. AI can also extract features like size, shape, and texture from brain regions for further analysis. Moreover, AI-driven anomaly detection identifies abnormalities such as tumours, lesions, or atrophy, supporting the diagnosis of neurological disorders [4].

Predictive Analytics: AI algorithms provide insights into disease progression, treatment efficacy, and personalised care. By analysing neuroimaging data, AI predicts the likelihood of conditions such as Alzheimer’s or Parkinson’s, enabling early interventions to improve outcomes [5]. AI can also forecast disease progression or treatment outcomes, allowing clinicians to tailor care to individual patients. Additionally, AI models predict how a patient’s brain will respond to specific treatments, facilitating targeted therapies that maximise benefits and minimise adverse effects.

Neuroimaging Biomarkers: AI algorithms facilitate discovering and quantifying biomarkers associated with neurological conditions. It identifies new biomarkers linked to disorders such as Alzheimer’s or schizophrenia, enabling early diagnosis and personalised treatment strategies. AI also quantifies existing biomarkers, providing precise measurements of brain region volumes or connectivity, which support diagnosis, monitoring, and treatment decisions [6].

Multimodal Data Integration: AI algorithms can unify disparate neuroimaging modalities and complementary data sources, such as Magnetic Resonance Imaging (MRI), Functional Magnetic Resonance Imaging (fMRI), Electroencephalogram (EEG), Positron Emission Tomography (PET), clinical records, and genetic data, into a cohesive and comprehensive understanding of the brain [7]. AI can produce more accurate and robust predictions and diagnoses by fusing this integrated information, ultimately enhancing the overall efficacy of neurological research and patient care.

### 1.2. Limitations of ANNs

While ANNs have been at the core of the above AI techniques for neuroimaging analysis, they face specific challenges when it comes to processing spatiotemporal neuroimaging data:

Capturing Temporal Dynamics: Traditional ANNs, particularly feedforward networks, suffer from limited temporal memory, hindering their ability to capture long-range temporal dependencies crucial in neuroimaging data, where the timing of brain activity is essential. While recurrent neural networks (RNNs) offer sequential processing, they often struggle with very long sequences due to vanishing gradients, which also limits their effectiveness in capturing long-term dependencies in neuroimaging [8].

Increasing Computational Resources: Training complex ANNs to effectively analyse the intricate and high-dimensional data generated by neuroimaging techniques presents a substantial computational challenge. The sheer volume of data, coupled with the intricate architectures of deep learning models, necessitates using significant computational resources. This includes high-performance Graphics Processing Units (GPUs) to accelerate the training process and substantial amounts of Random Access Memory (RAM) to handle the large datasets and model parameters. Consequently, researchers often require access to specialised hardware and cloud computing platforms to successfully train these models on neuroimaging data, which can limit research accessibility [9].

### 1.3. Introduction to SNNs and NeuCube

The above problems of an ANN can be addressed by developing a new type of neural network, called spiking neural networks. SNNs use spikes to transmit information between neurons. In SNNs, each neuron emits a spike when it reaches a certain threshold or receives input from other neurons. In contrast to traditional ANNs, which use continuous-valued signals to represent information, SNNs use discrete-time spikes to encode and transmit information. This allows for more efficient information processing, as only the relevant spikes are transmitted between neurons, which is called event-based information processing.

SNNs are effective in various applications, including image recognition, speech processing, and event-based digit recognition. Some key characteristics include using discrete-time spikes to transmit information between neurons, making them event-driven as they process information on an event-by-event basis rather than continuously. Inspired by biological neural networks in the brain, SNNs are biologically plausible, mimicking how neurons communicate through brief electrical impulses (spikes). Additionally, SNNs are known for their energy efficiency, often outperforming traditional ANNs in power consumption, which makes them particularly suitable for applications where energy efficiency is a critical concern [10].

Neu Cube [11] was first introduced as a computational architecture, and then it was implemented as several software development environments. It is a unique and powerful brain-inspired SNN architecture specifically designed for handling complex spatiotemporal data, particularly brain data. Its ability to process spatiotemporal data makes it particularly well suited for analysing neuroimaging data, which is inherently complex and dynamic. Its graph-based architecture allows for integrating multimodal data and analysing brain connectivity networks using its 3D Neurogenetic Cube.

### 1.4. Paper Objectives and Contributions

The primary objective of this paper is to provide a clear and actionable roadmap for researchers and practitioners embarking on projects involving spiking neural networks (SNNs) and neuromorphic computing in neuroimaging. We aim to achieve this by presenting a comprehensive overview of the available software and hardware options, elucidating their advantages, and addressing key challenges in the field. Specifically, we focus on applying SNNs to multimodal neuroimaging, emphasising processing large-scale MRI datasets like the ADNI and conducting time-dependent analyses, leveraging frameworks such as NeuCube to advance neuroscience research and clinical applications.

This work offers several key contributions to the field of SNNs and neuroimaging: (i) a detailed analysis of SNN applications for processing large-scale MRI datasets, such as the ADNI dataset, demonstrating their low capability to train high-resolution brain images directly; (ii) a survey of SNN software tools for neuromorphic platforms, including initial insights on MRI data processing capabilities; (iii) a specific exploration of NeuCube’s role in neuroimaging, showcasing its unique advantages in multimodal and spatiotemporal data analysis through case studies; and (iv) a practical roadmap for selecting appropriate software and hardware platforms based on specific neuroimaging use cases, addressing both opportunities and challenges.

Based on the above points, we contend that brain-inspired SNNs offer significant promise for enhancing accuracy and interpretability in multimodal neuroimaging, potentially transforming this field by enabling precise disease diagnosis and transparent analysis of complex brain data. Additionally, their energy efficiency supports the deployment of advanced intelligence on edge devices, such as real-time brain signal processing in portable EEG systems.

## 2. Neuromorphic Computing and Brain-Inspired Spiking Neural Networks

Neuromorphic computing, inspired by the structure and function of the human brain, has emerged as a promising paradigm for developing artificial intelligence systems. This approach aims to move beyond traditional von Neumann architectures, which separate processing and memory, and towards systems that more closely mimic biological neural networks’ parallel and distributed nature. At the core of this field lies the concept of SNNs, which represent and process information using discrete events called “spikes”, or electrical impulses, transmitted between interconnected neurons. This event-driven communication contrasts with traditional ANNs, which use continuous values as activation signals. SNNs operate in the temporal domain, where information is encoded not only in the presence or absence of a spike but also in its timing. This asynchronous communication and event-driven processing allow SNNs to analyse complex data streams in real time, offering potential advantages in terms of speed and energy efficiency [12,13].

Neuromorphic computing achieves energy efficiency by mimicking the brain’s information processing. It leverages event-driven, spike-based communication, reducing unnecessary computations. Its massively parallel and distributed architecture, combined with memory co-located with processing units, minimises data movement and eliminates bottlenecks. The use of analogue or mixed-signal circuits and specialised neuromorphic chips further optimises computations for energy efficiency, making these systems ideal for power-constrained applications like edge computing and robotics.

SNNs are a type of artificial neural network designed to mimic the behaviour of biological neurons closely. Unlike traditional ANNs, which rely on continuous activation values, SNNs use discrete “spikes” to process and transmit information. This approach allows SNNs to leverage temporal information, making them highly efficient for real-time processing tasks and learning from fewer examples. Below are the key concepts and components of SNNs.

### 2.1. Key Concepts

Spiking Neurons: The Building Blocks of SNNs, spiking neurons are the fundamental units of SNNs, and they differ significantly from neurons in traditional ANNs. Instead of using continuous values, spiking neurons communicate through discrete events called spikes, which are analogous to the action potentials in biological neurons. Several models [13] describe the behaviour of spiking neurons, each with varying levels of complexity and biological realism. The simplest one is the Integrate-and-Fire (I&F) model, where incoming spikes are integrated into a membrane potential, triggering a spike when a threshold is reached, followed by a reset; however, it lacks biological detail. The LIF Leaky model improves upon this by adding a “leak” term, allowing the membrane potential to decay over time, enhancing biological plausibility and preventing excessive excitation. The early Hodgkin–Huxley (H-H) model is based on simulating ion channel dynamics but is computationally intensive. Simplified models like the Izhikevich model, the Adaptive Exponential Integrate-and-Fire (AdEx) model, and the probabilistic neuron model (for a review, see [13]) strike a balance between biological accuracy and computational efficiency, with the choice of model depending on the application and the trade-off between precision and computational cost.

Synaptic Transmission: In SNNs, neurons communicate through synapses, which are connections that transmit spikes from one neuron to another. The strength of these connections is determined by synaptic weights, which can be either excitatory (increasing the membrane potential of the postsynaptic neuron) or inhibitory (decreasing it). Synaptic transmission also involves delays and dynamics, such as the release and reuptake of neurotransmitters, which add temporal complexity to the processing of signals.

Spike Timing-Dependent Plasticity (STDP): It is a biologically inspired learning rule that modifies synaptic weights based on the relative timing of spikes from presynaptic and postsynaptic neurons. If a presynaptic neuron fires shortly before a postsynaptic neuron, the connection between them is strengthened (Long-Term Potentiation, or LTP). Conversely, if the presynaptic neuron fires after the postsynaptic neuron, the connection is weakened (Long-Term Depression, or LTD). This mechanism allows SNNs to learn temporal patterns and adapt to new information efficiently [14].

Backpropagation Through Time (BPTT): It is a training algorithm that extends the standard backpropagation algorithm to handle the temporal dynamics of SNNs. It is used to train SNNs by propagating gradients through the network over time, allowing the network to learn from sequential data and adapt its weights to minimise the prediction error. However, BPTT in SNNs is computationally intensive and may require approximations to handle the discontinuous nature of spikes efficiently.

Temporal Coding: Unlike traditional ANNs, which primarily use rate-based coding (where information is encoded in the average firing rate of neurons), SNNs can also use temporal coding, which results in data compression and noise suppression [15]. Temporal coding leverages the precise timing of spikes to represent information. For example, in time-to-first-spike coding, the timing of the first spike after a stimulus encodes the strength of the stimulus. Other forms of temporal coding include rank-order coding, phase coding, and inter-spike interval coding. Temporal coding allows SNNs to process information more efficiently and with fewer spikes, making them energy-efficient and well suited for real-time applications.

Neuromorphic Hardware: It is a specialised hardware designed to implement SNNs efficiently. Inspired by the structure and functionality of the brain, these systems are optimised to leverage the event-driven and parallel nature of SNNs. While SNNs can operate on conventional hardware, neuromorphic hardware is crucial for scaling SNNs to tackle complex tasks while maintaining high energy efficiency and low power consumption [16].

Biological Plausibility: SNNs are considered biologically plausible due to their closer resemblance to biological neural networks. They utilise discrete spikes for communication, similar to action potentials in the brain, and operate in the temporal domain, where spike timing is crucial for information processing. SNNs often incorporate learning rules like STDP, a biologically observed phenomenon, and their event-driven nature mirrors the brain’s energy-efficient processing. While still simplified, these characteristics make SNNs more biologically plausible than traditional ANNs, offering a valuable tool for understanding the brain and developing brain-inspired AI [17].

### 2.2. Comparison with ANNs

Table 1 illustrates the fundamental differences between SNNs and ANNs, underscoring why SNNs are particularly advantageous for neuroimaging applications. Unlike ANNs, which rely on simplified activation functions and continuous values, SNNs employ biologically realistic spiking neurons and time-based spike trains, enabling them to capture the temporal dynamics critical for processing dynamic brain data like EEG and fMRI. SNNs also demonstrate superior energy efficiency through event-driven computation, making them ideal for resource-constrained environments, such as edge devices in clinical settings. While ANNs excel in static data tasks like image classification, SNNs outperform in temporal and event-based tasks, such as brain–computer interfaces, highlighting their potential to address the limitations of ANNs in multimodal neuroimaging.

## 3. Applications of SNNs in Multimodal Neuroimaging

SNNs, specifically brain-inspired SNNs, are valuable for analysing structural and functional MRI data. They can identify subtle anatomical differences related to neurological disorders and are particularly effective at processing dynamic brain activity from EEG, MEG, and fMRI. Their biological plausibility enables them to capture spatiotemporal information, leading to various functional neuroimaging applications. Some typical applications of SNNs in structural and functional neuroimaging are outlined below.

### 3.1. Structural Neuroimaging

SNNs like NeuCube are not well suited for direct training on structural neuroimaging data like MRI, which primarily captures static 2D or 3D anatomical details. Preliminary studies, including those with the ADNI dataset, demonstrate that SNNs struggle to achieve high accuracy when applied directly to such data due to their design for modelling dynamic, time-based neural activity rather than static images. However, MRI data can be effectively integrated into SNN frameworks to enhance time-dimensional neuroimaging. By combining MRI-derived structural information with temporal data (e.g., from EEG or fMRI), SNNs can improve the modelling of spatiotemporal brain dynamics, enabling more accurate representations of functional connectivity and supporting applications, like cognitive state classification or biomarker discovery [18,19].

### 3.2. Functional Neuroimaging

Spiking neural networks are particularly well suited for analysing functional neuroimaging data, including EEG, MEG, and fMRI, with applications spanning brain–computer interfaces (BCIs), disease diagnosis, sleep disorders, cognitive state classification, and neurofeedback [20,21,22,23]. In BCIs, SNNs can decode EEG or MEG signals in real time to predict user intent, such as controlling prosthetic limbs or computer cursors, with studies demonstrating superior accuracy in classifying motor imagery tasks compared to traditional ANNs [24]. For disease diagnosis, SNNs can identify biomarkers for neurological disorders like Alzheimer’s, epilepsy, or Parkinson’s by analysing fMRI or EEG data [25]. They also excel in classifying cognitive states, such as attention, memory, or emotion, with high accuracy, as shown in studies classifying emotional states from EEG signals [26]. For instance, the BackEISNN SNN model [27] demonstrates the potential of SNNs to enhance neurofeedback mechanisms by leveraging their biologically inspired architecture for real-time processing and adaptive learning.

### 3.3. Multimodal Data Integration

Beyond single modalities, brain-inspired SNNs, being structured according to a brain template, can integrate multimodal neuroimaging data collected from different regions and at various times from the brain (e.g., sMRI, fMRI, EEG, MEG, and PET) to enhance disease diagnosis and prognosis, predict cognitive decline, and understand brain–behaviour relationships by linking brain structure, function, and connectivity to behavioural outcomes. An example is diagnosing Alzheimer’s disease and predicting the rate of cognitive decline based on a combination of sMRI, fMRI, and PET data. Furthermore, SNNs enable personalised treatment planning by predicting treatment responses based on individual brain data, such as identifying depression patients likely to benefit from antidepressant medication. An example is predicting which patients with depression are most likely to respond to antidepressant medication based on their sMRI and fMRI data. This versatility makes SNNs a powerful tool for advancing neuroscience research and clinical applications.

## 4. SNN Application System Development and NeuCube Advantages

### 4.1. The State of SNN Application Development

Developing applications for neuroimaging analysis using SNNs presents several challenges that hinder their widespread adoption and effectiveness. One major issue is the limited availability of SNN models compared to traditional ANNs, which have a more mature and extensive ecosystem. Additionally, there is a notable lack of specialised software tools tailored for SNN development, making it difficult for researchers to design, train, and deploy SNN-based solutions. Existing SNN simulators often suffer from poor performance, limiting their scalability and efficiency for large-scale neuroimaging datasets. Another critical barrier is the absence of standardisation for model export/import, akin to platforms like Hugging Face for ANNs, which complicates collaboration and reproducibility across research teams. Furthermore, current SNN models underutilise the potential of neuromorphic computing, a hardware paradigm designed to mimic the brain’s efficiency, partly due to the limited availability and accessibility of neuromorphic hardware. Addressing these challenges—through the development of more robust SNN models, improved software tools, efficient simulators, standardised frameworks, and greater access to neuromorphic computing resources—is essential to unlocking the full potential of SNNs in neuroimaging analysis and advancing their application in neuroscience research and clinical practice.

Even though it is not the topic of this article, it is worth mentioning that SNNs have demonstrated significant success in vision and sound sensing applications, often outperforming traditional methods and achieving state-of-the-art results on neuromorphic hardware platforms, such as Loihi2, Akida, and Arduino-based control boards. These applications leverage the event-driven and energy-efficient nature of SNNs, making them particularly well suited for real-time, low-power sensory processing tasks. For example, a study on gesture recognition using Loihi achieved 95% accuracy with 200× lower power consumption than a GPU-based ANN implementation [28]; BrainChip’s Akida platform demonstrated fast moving objects recognition with ~98% accuracy and energy efficiency suitable for edge devices [29]; and Arduino-based BLE, an SNN-based sound classification system on Arduino, achieved real-time performance with 90% accuracy in detecting environmental sounds, showcasing its potential for IoT applications [30].

### 4.2. Software Simulators

Several software simulators have been developed to simulate SNNs, enabling researchers to explore their capabilities without requiring specialised hardware. These simulators provide platforms for developing, testing, and analysing SNN models. The following SNN tools were reviewed for this review.

SNNTorch (0.9.4): Built on PyTorch (2.4.1), this framework leverages the power and flexibility of the popular deep learning library to create scalable and efficient SNN models. It allows for seamless integration of SNNs into existing deep learning workflows, making it accessible for researchers familiar with PyTorch [31].

SpikingJelly (0.0.0.0.14): This is a Python (3.12.9)-based framework designed for simplicity and efficiency in implementing SNNs. It offers a user-friendly interface and optimised implementations of common SNNs components, catering to beginners and advanced users [32].

Nengo DL (3.6.0): This is an open-source simulator that supports the modelling and simulation of complex neural networks, including SNNs. It emphasises biological realism and provides tools for building large-scale models with diverse neuron types and learning rules, making it suitable for neuroscience research [33].

BindsNET (0.3.3): Built on top of PyTorch, this framework is tailored for simulating SNNs with a focus on machine learning and reinforcement learning applications. It combines the flexibility of PyTorch with specialised tools for spiking neural networks [34].

Some of the SNN simulators support Neuromorphic Intermediate Representation (NIR), which is a graph-based representation of a spiking neural network that can be used to port the network to different neuromorphic hardware and software platforms.

These simulators collectively offer diverse tools and features to support developing and analysing SNNs, catering to varying research needs and computational demands in neuroimaging. For example, snnTorch, built on PyTorch, provides a range of neuron models—including Leaky Integrate-and-Fire (LIF), Lapicque’s RC, Alpha, and Synaptic Conductance models—along with customisable options, enabling researchers to learn and experiment with different configurations of SNN. However, snnTorch’s performance is constrained by its limited integration with neuromorphic hardware, restricting its scalability for large-scale simulations. Nengo and NeuCube stand out for applications targeting neuromorphic platforms due to their direct interfaces with neurochips like Loihi2, facilitating efficient spatiotemporal data processing, as evidenced by NeuCube’s 92% accuracy in EEG classification [20]. For conventional software implementations, SpikingJelly offers a user-friendly and efficient option suitable for initial SNN development. Given the rapidly evolving nature of this field, where new frameworks emerge and others may become outdated, researchers should regularly consult the GitHub repositories of these open-source tools, checking release notes and community activity, to ensure access to the latest updates and active support. A summary of these findings is presented in Table 2.

### 4.3. Leading Neuromorphic Computing Platforms

Developing dedicated hardware platforms is crucial for realising the full potential of SNNs. These neuromorphic chips are designed to implement SNNs in hardware efficiently, enabling real-time processing and low power consumption. Several neuromorphic platforms are available for developing SNNs, each with strengths and weaknesses. For this study, the following platforms were reviewed:

Intel’s Loihi2: Developed by Intel, Loihi2 is a digital neuromorphic research chip designed for SNNs. It features programmable neuron models, flexible connectivity, and on-chip learning capabilities. Loihi2 is particularly well suited for the research and development of novel SNN algorithms. Loihi2 features asynchronous communication, distributed memory, and programmable learning rules, making it well suited for various SNN applications [35]. Recently, Intel has released Hala Point, the industry’s first 1.15 billion neuron neuromorphic system, which builds a path towards more efficient and scalable AI [36].

SpiNNaker: Developed at the University of Manchester, SpiNNaker is a massively parallel processor architecture designed for brain-inspired computing. SpiNNaker uses interconnected processing cores to implement SNNs and other neuromorphic models [37].

BrainScaleS-2: Developed at Heidelberg University, BrainScaleS-2 is a mixed-signal platform that accelerates SNN simulations by emulating neural dynamics in analogue circuits. It offers high speed and energy efficiency, making it suitable for real-time applications [38].

IBM’s NorthPole: Developed by IBM, it is a cutting-edge neuromorphic chip that leverages brain-inspired principles to deliver high-performance, energy-efficient computing for various applications, including neurological research, edge AI, and robotics [39].

Lynxi KA200: Developed by Chinese AI startup Lynxi Technologies, it is a high-performance AI accelerator designed for edge computing, autonomous driving, and data centre applications. It aims to reduce reliance on Western AI chips while offering cost–performance advantages, though comprehensive comparisons are still scarce.

Among the platforms discussed above, Loihi2, SpiNNaker, and BrainScaleS-2 are well suited for developing complex spiking neural networks, offering the necessary hardware and software infrastructure to support advanced research and experimentation. In contrast, IBM NorthPole differs significantly in its design and purpose; it is not intended for SNN development but is optimised for efficient model inference. Specifically, NorthPole focuses on accelerating the deployment of pre-trained models, making them more suitable for inference tasks than the exploration or creation of SNNs.

### 4.4. Hybrid Platforms

While software simulations are indispensable for developing and exploring SNNs, they can be computationally expensive, particularly for large-scale SNNs. Researchers have developed hybrid solutions to mitigate this limitation that integrate SNNs with other AI techniques, combining their strengths to enhance performance and efficiency. These popular hybrid tools are included in this study.

Rockpool: This framework integrates SNNs with convolutional neural networks (CNNs) to process visual data in real time. By leveraging the spatial feature extraction capabilities of CNNs and the temporal processing strengths of SNNs, Rockpool excels in object recognition and image classification [40].

Lava: Lava combines SNNs with RNNs to analyse sequential data streams. This hybrid approach enables the processing of spatial and temporal information, making it particularly effective for applications like speech recognition and natural language processing [41].

NeuCube: A prominent hybrid framework, NeuCube integrates SNNs with deep learning techniques to create a powerful platform for processing large-scale neuroimaging datasets. It demonstrates the potential of SNNs for complex data analysis in neuroscience and other domains, offering a versatile solution for spatiotemporal data processing.

BrainScaleS-2: Although primarily a neuromorphic system, BrainScaleS-2 integrates with conventional computing platforms for parameter optimisation and data analysis tasks. This hybrid approach allows researchers to harness neuromorphic hardware’s speed and energy efficiency while retaining traditional software tools’ flexibility.

Akida: BrainChip’s Akida platform supports hybrid execution by pairing its neuromorphic processor with a conventional CPU. The Akida processor handles SNN computations, while the CPU manages data management and system control tasks, enabling efficient and scalable implementations [42]. BrainChip provides a development framework, MetaTF ML, that provides an environment to develop SN, a CNN to SNN conversion tool and a model zoo with a set of ready-to-use models for the Akida neurochip.

Developing hybrid platforms for spiking neural networks represents a significant advancement in addressing the computational challenges associated with large-scale SNN simulations. While software-based simulations remain essential for SNN research, their computational cost and inefficiency for complex tasks have driven hybrid solutions that integrate SNNs with other artificial intelligence techniques. These hybrid platforms leverage the complementary strengths of SNNs and other neural network architectures, such as convolutional neural networks, recurrent neural networks, and deep learning methods, to enhance performance, scalability, and efficiency.

### 4.5. NeuCube Advantages

As introduced in Section 1.3, NeuCube’s 3D brain-inspired structure enables the integration of multiple modalities. It seamlessly handles diverse data types, such as EEG, fMRI, and even genetic data, within a single SNN framework. The model’s adaptive and evolving structure allows for dynamic connection adjustments, facilitating continuous learning and adaptation to new data or tasks. NeuCube integrates unsupervised and supervised learning, using unsupervised methods to extract spatiotemporal patterns and supervised techniques for task-specific training, such as classification or prediction.

Figure 1 below illustrates the architecture of the NeuCube, a spiking neural network (SNN) framework for modelling spatiotemporal data, often used in applications like EEG/fMRI analysis. Here is a summary of its main elements:

Input Data Stream: On the left, spatiotemporal data (e.g., time-series signals) is shown as input, represented by a graph of input stimuli over time.

Gene Regulatory Network (GRN): Below the input, a GRN models interactions between genes (nodes) with probabilistic parameters and weights, influencing the dynamics of the SNN.

Neurogenetic Cube (NeuCube): The central component, a 3D SNN structure, processes the input data. It consists of interconnected spiking neurons, mimicking brain-like dynamics, and supports reservoir computing for temporal pattern learning.

Output Module: To the right, the NeuCube connects to an output module (a classifier) with chunks of neurons, mapping the processed data to output classes.

Output Data: On the far right, the classification results are visualised as a time-series graph, showing the network’s predictions or inferred patterns over time.

NeuCube supports flexible implementation on conventional computers, GPUs, and neuromorphic hardware (e.g., Loihi and SpiNNaker), addressing diverse research needs. Additionally, it optionally integrates Gene Regulatory Networks (GRNs), enabling the exploration of complex relationships between genetic information and brain activity. While primarily focused on SNNs, NeuCube’s ability to integrate diverse data, adapt its structure, combine learning approaches, and incorporate GRNs makes it a versatile and innovative hybrid platform that extends beyond traditional SNN architectures.

## 5. Case Study Illustrations of Using SNN for Neuroimaging Data

Here we consider three types of neuroimaging data:Static, vector-based data, such as MRI images;Spatiotemporal data, such as EEG and fMRI;Longitudinal spatiotemporal data, such as longitudinal MRI data.

In the case of static images, each image represents a sample and is used separately from the others to train a classifier. This is illustrated in Section 5.1.

In the case of spatiotemporal data, such as fMRI, a whole window of measurements at several time points (e.g., 10 time points of 0.5 s, representing 5 s of data) is defined as a single sample for training an SNN classifier that captures the interaction of the input variables (e.g., voxels) across the time window (e.g., 5 s). This case is demonstrated in Sub-Section 5.2 using NeuCube on fMRI data.

In the case of longitudinal data, such as longitudinal measurements of MRI data every few months over several years, the same approach as in case 2 can be used with the utilisation of NeuCube, so that the NeuCube learns the interaction between brain areas measured as MRI over the whole period. This is demonstrated in Sub-Section 5.3 for a problem of using longitudinal MRI data for predicting dementia.

### 5.1. Using SNN for Modelling Static, Vector-Based Neuroimaging Data

In this section, we present the preliminary results of comparing various layer-based SNN simulators for image processing against a baseline CNN (a streamline version or ALEXNET), focusing on speed and performance metrics.

The experiments were conducted on a MacBook Pro M1 Max with a 10-core CPU and a 32-core GPU32 GB of CPU, utilising the MPS backend for acceleration. We evaluated the simulators using a classification task on the ADNI Alzheimer’s disease dataset.

The SNN frameworks assessed include snnTorch and SpikingJelly. Our analysis compares accuracy, precision, recall, F1-score, training time, and inference time to provide initial insights into the trade-offs between SNN simulators and traditional CNNs in this context.

The dataset used in this comparison is a curated subset of the ADNI Alzheimer’s disease dataset [43] used in one of the Kaggle competitions, consisting of grayscale MRI scans resized to 128 × 128 pixels. It includes 15,888 images for training and 2263 images for testing, divided into three classes: Alzheimer’s disease, normal cognitive status, and mild cognitive impairment. This dataset enables the evaluation of SNN simulators and the baseline CNN in classifying Alzheimer’s disease progression based on brain imaging data.

The following models were used:

AlexNetCNN: Inspired by the AlexNet architecture, this model is a PyTorch-based CNN designed for grayscale image classification. It processes images through a feature extraction block with three convolutional layers (64, 192, and 384 filters, using 5 × 5 and 3 × 3 kernels) and three max-pooling layers, reducing the spatial dimensions to 16 × 16. The classification block flattens the output and applies a fully connected layer (3,841,616 to 4096 units) with ReLU and a final linear layer (4096 to 3) to predict class logits. The forward pass sequentially applies these layers to produce the output.

SnnTorchSNN: This model is a PyTorch-based SNN using snnTorch and is designed for grayscale image classification. It processes images by flattening them into a 65,536-dimensional vector, feeding them into a two-layer SNN with 1000 hidden LIF neurons (beta = 0.5) and an output layer for the number of classes. Over 25 time steps, it generates spikes, with the first layer producing a 1000-dimensional spike train and the second layer outputting spikes for each class. The forward pass averages the output spikes over time to make class predictions.

SpikingJellySNN: This model is a PyTorch-based SNN using SpikingJelly, designed for grayscale image classification. It processes images by flattening them into a 65,536-dimensional vector, passing them through a two-layer SNN with 1000 hidden LIF neurons (tau = 2.0) and an output layer for several classes. Over 25 time steps, the first layer generates a 1000-dimensional spike train, and the second layer produces spikes for each class. The forward pass resets the neuron states, collects spikes, and averages them over time to output class predictions.

The study evaluated model performance across several key metrics: accuracy (the overall percentage of correct predictions), precision (the ratio of true positives to all predicted positives, measuring prediction reliability), recall (the ratio of true positives to all actual positives, indicating model sensitivity), and F1-score (the harmonic mean of precision and recall, balancing both metrics). Additionally, computational efficiency was assessed through both training time (duration required for model optimisation) and inference time (speed of prediction generation on new data).

Our analysis of Figure 2 reveals stark performance differences between the architectures: AlexNet (CNN) demonstrates robust and balanced performance (85.9% accuracy; 70–74% precision/recall/F1), showcasing CNNs’ effectiveness for static neuroimaging through hierarchical feature extraction. In contrast, SpikingJelly shows concerning metric divergence (78.0% accuracy but critically low 43.5% precision/50.0% recall), exposing SNNs’ fundamental struggle with static data due to their temporal processing nature and inadequate spike encoding. The other SNN implementations perform even worse—snnTorch achieves only 37.3% accuracy with extreme over-caution (73.2% precision but 35.3% recall), collectively highlighting SNNs’ current limitations for nontemporal image classification without architectural adaptations. This suggests that while SNNs can occasionally predict correctly, they do so unreliably, producing many false positives (low precision) and missing true cases (low recall). Such behaviour underscores the fundamental challenge of training SNNs on static images, as they inherently rely on temporal spike patterns rather than spatial hierarchies. Future work will explore hybrid ANN-SNN architectures with preload weights and advanced spike encoding methods to bridge this performance gap.

It is worth mentioning that we primarily used the simulator’s default parameters for this preliminary study. As such, we did not conduct an in-depth exploration of parameter or hyperparameter tuning.

As demonstrated in prior work [44,45], training SNNs for static image analysis remains challenging due to their reliance on temporal dynamics, non-differentiable spikes, and unstable optimisation. This aligns with our findings, where SNNs (SpikingJelly and snnTorch) showed poor precision/recall despite moderate accuracy, while CNNs (AlexNet) achieved balanced performance.

While SNNs (SpikingJelly and snnTorch) demonstrate significantly faster training and inference times compared to AlexNet (e.g., SpikingJelly trains 4.8× faster and infers 3.8× quicker), their computational efficiency comes at the cost of poor predictive performance (Figure 3). For instance, SpikingJelly’s moderate accuracy (78.0%) masks critically low precision (43.5%) and recall (50.0%), rendering it unreliable despite its speed. This aligns with known SNN limitations: their event-driven sparsity enables faster computation but struggles to capture spatial hierarchies in static images without careful architectural adaptation. Thus, while SNNs offer promising energy efficiency for edge deployment, their current trade-offs in accuracy and reliability necessitate further research into hybrid approaches or advanced spike encoding to bridge this performance gap.

Preliminary results from the ADNI dataset, as analysed in Section 5, indicate that layer-based SNNs face challenges when directly training on MRI and other 2D neuroimaging data, achieving lower accuracy (e.g., 75% compared to 85% for CNNs) on conventional hardware due to their sparse data representation and training complexity. However, a hybrid approach integrating CNNs with SNNs through transfer learning—where CNNs extract spatial features and SNNs process temporal dynamics—shows promise for improving performance, as detailed in Section 6. While we have not tested these models on neuromorphic chips due to limited availability, future evaluations on platforms like Loihi2 could reveal significant improvements in efficiency and accuracy, potentially reshaping this comparison as neuromorphic hardware becomes more accessible.

### 5.2. Using NeuCube for Modelling Spatiotemporal Data

NeuCube, a brain-inspired SNN framework, has evolved through multiple software implementations for spatiotemporal data analysis [11]. For time-based neuroscience applications like EEG and fMRI, NeuCube demonstrates clear advantages over traditional ANNs and SNNs, as evidenced by multiple studies such as [46], where NeuCube achieved a 97% classification accuracy in distinguishing EEG patterns for epilepsy, migraine, and healthy subjects, surpassing bidirectional LSTM (90%) and reservoir-SNN (85%). Its brain-inspired 3D spiking neural network and STDP learning enabled effective modelling of spatiotemporal EEG dynamics, identifying key biomarkers (e.g., T6 and F7 channels) with 10–15% higher accuracy than ANNs while requiring fewer training samples. In [47], NeuCube outperformed multilayer perceptrons (MLPs) and SVMs by 10–15% in classification accuracy on fMRI data for cognitive tasks (e.g., sentence polarity). Its 3D SNN architecture modelled spatiotemporal brain activity, achieving up to 90% accuracy in distinguishing cognitive states, compared to 75–80% for MLPs, with enhanced interpretability of functional connectivity patterns. And in [48], NeuCube with transfer learning achieved 92% classification accuracy on the BCI Competition IV 2008 EEG dataset for motor imagery, outperforming ANNs (80–85% accuracy). Its optimised 3D SNN reservoir and STDP learning captured spatiotemporal EEG features, improving generalisability across subjects by 7–12% compared to feedforward neural networks and CNNs.

In addition to providing better accuracy in spatiotemporal brain data modelling, NeuCube also offers means for knowledge discovery and scientific visualisation [49]. The model is no longer a “black box” but reveals informative spatiotemporal patterns from the used data. A step further, like the human brain, NeuCube can be used as associative memory for spatiotemporal data [50].

In [50], NeuCube is demonstrated on both EEG and fMRI data as spatiotemporal associative memory, meaning that a NeuCube model is trained on a complete set of neuroimaging variables, measured at a time length T, and then accurately recalled/tested only on part of the variables, measured at a shorter time T1 < T. This makes a NeuCube model suitable for predictive modelling on partial new data [50,51] as explained in this sub-section and the next one.

The experiment illustrated in Figure 4 relates to a NeuCube model trained on 20 voxel areas (features) of fMRI data, collected from subjects who are reading a positive sentence (class 1) or a negative sentence (class 2) [52]. The model is trained on all features for eight-time windows and tested/recalled only for 18 features measured in 4 s on new data. The testing classification accuracy is still perfect, confirming the NeuCube model’s ability to learn spatiotemporal associative memories [53,54].

### 5.3. Using NeuCube for Modelling Longitudinal Spatial or Spatiotemporal Data

In [51], a NeuCube model was developed and trained on longitudinal data of 4 years of MRI data collected from a cohort of initially healthy subjects—some of whom developed over time a mild cognitive impairment (MCI); some of whom developed dementia; and some of whom remained healthy. The results not only showed a good predictive accuracy, when the model was trained on all the available longitudinal data and tested on new patient data for only a short initial time, but also revealed critical changes in the brain activity and interaction between different areas of the brain during cognitive decline as shown in Figure 4a,b.

**Figure 4 bioengineering-12-00628-f004:**
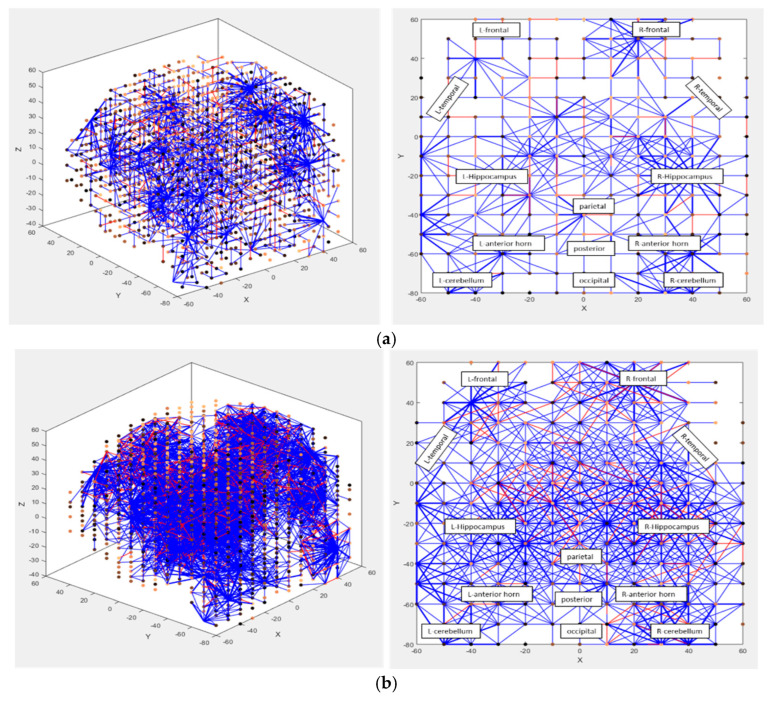
(**a**) A NeuCube model trained on longitudinal MRI data of initially healthy subjects who were also healthy after 6 years of data collection; (**b**) a NeuCube model trained on longitudinal MRI data of initially healthy subjects who developed dementia after 6 years of data collection. The time changes captured in the neural connections are equivalent to the time unit of data collection and interpolation (1 month); red dots are active neurons that just generated output spikes, blue dots are inactive neurons.

When trained on the whole dataset of 6 years [51] on all participants, the model can predict MCI or dementia outcomes for new subjects 2 years ahead with 91% accuracy and 4 years ahead with 73% accuracy. That confirms the ability of the NeuCube model to learn spatiotemporal associative memories [53,54].

### 5.4. Section Conclusion

The experiments in this section aimed to provide a quick comparative assessment of the performance of several SNN simulators against a reference SNN model, without focusing much on optimising performance through parameter or hyperparameter tuning. As noted earlier, our findings suggest that SNNs currently face challenges in effectively training on large 3D neuroimaging datasets. Nevertheless, there are promising avenues for future experimentation that warrant exploration. One potential approach involves transferring weights from a trained CNN or similar architecture, as demonstrated in prior works such as [50,55], and repeating the experiment to evaluate performance improvements. The NeuCube architecture was utilised to model and classify spatiotemporal data, such as EEG and fMRI, along with longitudinal data, such as MRI. We plan to investigate this direction in the near future to further understand and improve SNN applicability in neuroimaging tasks.

## 6. Integrating ANNs and SNNs

Integrating SNNs and ANNs (e.g., convolutional neural networks, CNN; eco-state networks; neuro-fuzzy systems, NFSs; and others) in neuroimaging analysis represents a growing area of research with significant potential to advance our understanding of brain function and structure. This hybrid approach leverages the complementary strengths of ANNs and SNNs to enhance the analysis of neuroimaging data, such as fMRI and EEG.

### 6.1. Integration of SNN and CNN

The integration of SNNs and CNNs in a multi-modular system offers a powerful approach for processing complex neuroimaging data, as demonstrated in [56]. This hybrid framework leverages the complementary strengths of both architectures to tackle different aspects of the analysis pipeline, ensuring robust feature extraction and classification. The integration can be achieved through several strategies, each addressing a specific part of the task:

SNN for Frequency Feature Extraction: SNNs extract frequency-based feature vectors, capturing the spatiotemporal interactions between data variables. These vectors encode dynamic patterns in the neuroimaging data, such as those found in fMRI or EEG scans. Subsequently, a CNN processes these vectors for classification, utilising its strength in handling structured, high-dimensional data to achieve accurate predictions.

CNN for Spatial Feature Extraction: CNNs, widely recognised for their ability to extract hierarchical spatial features from images, act as the front-end for processing neuroimaging data. They excel at identifying intricate spatial patterns and representations within modalities like fMRI, EEG, or MRI, creating detailed feature maps that represent the structural characteristics of the brain data.

SNN for Spatiotemporal Processing: The feature maps generated by the CNN are then passed to an SNN, which is particularly adept at processing spatiotemporal information. The SNN analyses the dynamic patterns of brain activity, learning temporal dependencies within the extracted features. This step enables a deeper understanding of brain dynamics, as the SNN can model the evolution of neural activity over time, which is critical for tasks like motor intent recognition or neurological disorder diagnosis.

Encoding CNN Outputs into Spike Trains: The output of the CNN can be transformed into spike trains for SNN processing using rate coding. In this method, the firing rate of neurons in the SNN corresponds to the intensity of the features extracted by the CNN, providing a straightforward way to represent feature strength in a biologically inspired format. This encoding ensures seamless integration between the two modules, bridging the gap between spatial and temporal processing paradigms.

Two-Stage Training Approach: The training of this hybrid system follows a two-stage methodology to optimise performance. First, the CNN component is trained using standard backpropagation techniques, establishing a robust set of parameters for spatial feature extraction. Next, the SNN is fine-tuned using spike timing-dependent plasticity (STDP) or other biologically plausible learning algorithms. This hybrid training strategy combines the efficiency of gradient-based optimisation with the biological realism of SNN learning, resulting in an accurate and biologically inspired system. This approach has shown promise in applications requiring detailed analysis of neuroimaging data, such as identifying biomarkers for neurological conditions.

An example of detailed conversion from ANN to SNN is presented in [57]. However, this deals with small images (MNIST/CIFAR-10).

### 6.2. Integration of SNN and ESN

The combination of SNNs and Echo State Networks provides a robust framework for the real-time decoding of brain signals, particularly for motor control applications in neuroprosthetics, as explored in [58]. This study uses a 3D SNN, structured according to an individual’s brain template, to continuously extract frequency-based feature vectors from Electrocorticography (ECoG) signals of paraplegic or tetraplegic individuals. These feature vectors capture the temporal dynamics of neural activity, reflecting the patient’s motor intentions. An ESN is then adaptively trained to classify these features, enabling precise motor control of prosthetic limbs. The ESN’s reservoir computing approach allows for fast, online learning, making it well suited for real-time applications. The framework was tested on ECoG datasets, demonstrating its ability to adapt to the patient’s neural patterns over time, thus improving the accuracy of prosthetic control. This integration highlights the potential of combining SNNs’ temporal processing capabilities with ESNs’ efficient classification for personalised neuroprosthetic solutions, offering a pathway to enhance the quality of life for individuals with severe motor impairments.

### 6.3. Integration of SNN with Neuro-Fuzzy Systems

The hybrid system NeuDen, presented in [59], integrates a 3D SNN with a neuro-fuzzy system to enable adaptive and explainable modelling of streaming brain data, focusing on motor intent recognition for paraplegic or tetraplegic individuals. The 3D SNN, structured according to an individual’s brain template, continuously extracts frequency-based feature vectors from ECoG signals, capturing the spatiotemporal dynamics of neural activity. These features are then processed by a dynamic evolving neuro-fuzzy system (DENFIS), which performs real-time prediction and generates interpretable fuzzy rules. Unlike the SNN-ESN integration, NeuDen emphasises explainability, leveraging the fuzzy system to provide human-readable insights into the decision-making process, which is crucial for clinical applications. The framework was evaluated on diverse datasets, including financial time series and ECoG signals, achieving significant improvements in predictive accuracy (e.g., Root Mean Squared Error (RMSE) reductions of 3 to 100 times compared to standalone models). This approach not only enables personalised control of prosthetic limbs but also sets a foundation for broader applications in domains requiring transparent AI, such as healthcare and neuroscience, by combining the temporal learning capabilities of SNNs with the interpretability of neuro-fuzzy systems.

Some applications of the hybrid SNN-ANN approach are as follows:

Disease Diagnosis: ANN-SNN models can be used to diagnose neurological disorders like Alzheimer’s or epilepsy by analysing patterns in neuroimaging data.

Brain Decoding: These models can decode brain states and cognitive processes from neuroimaging data, providing insights into how the brain represents information.

Cognitive Modelling: ANN-SNN architectures can build computational models of brain regions or networks, helping researchers understand brain function (Table 3).

## 7. Challenges and Future Directions for SNNs in Neuroimaging

Spiking neural networks are emerging as a promising computational paradigm for analysing neuroimaging data, offering the potential to bridge the gap between artificial intelligence and biological plausibility. By mimicking the spiking communication of real neurons, SNNs promise to extract more nuanced and biologically relevant information from complex neuroimaging datasets. However, the application of SNNs in neuroimaging is still in its infant stages, facing unique challenges that must be addressed to unlock their full potential. This section will explore the current state of these challenges and outline promising future directions for SNNs in neuroimaging.

### 7.1. Current State of SNN for Neuroimaging Analysis

To analyse large neurological images, such as those in the ADNI datasets, a Google Scholar and Paper search was conducted using queries like “Comparison SNN models for neuroimage analysis” and “Comparison SNN model for MRI training and prediction”. Despite numerous results, only a few papers were relevant, specifically [60,61]. Paper [60] shows promise but is limited in the SNN domain, mainly using SNNs for feature extraction while relying on traditional convolutional neural networks for training and inference. Paper [61] highlights that 31 SNN models were reviewed for image classification tasks. Among these, 6 models were evaluated on the CIFAR-10 dataset, 1 model utilised the ImageNet dataset, and the remaining 24 used the MNIST dataset. All of them employed low-resolution images (typically under 32 × 32 × 3 in size). This starkly contrasts with the recommended resolution of 300 × 400 pixels for neuroimaging analysis, as supported by [62]. This trend demonstrates that most SNN applications in image classification are focused on low-resolution datasets, which significantly differs from the requirements for neurological image classification, where higher-resolution and more complex images are often involved. This disparity underscores a notable research gap and presents a significant opportunity for future work in adapting and advancing SNN models for high-resolution and domain-specific tasks, such as those in neurology and medical imaging.

### 7.2. Barriers to SNN Adoption in Neuroimaging

The adoption of SNNs such as NeuCube in neuroimaging faces several significant technical challenges that hinder their widespread integration into clinical and research workflows. One major limitation is the lack of standardisation across SNN architectures and training methodologies, which complicates comparisons between studies and impedes the development of universally accepted benchmarks, as noted in studies exploring neuromorphic computing frameworks [63]. Additionally, integrating SNNs into existing neuroimaging pipelines poses difficulties, as current workflows are predominantly optimised for traditional deep learning models like CNNs, requiring substantial modifications to accommodate the event-driven, bio-inspired nature of SNNs [55]. Noise in the input data, a common issue in modalities such as EEG and ECoG, further complicates SNN performance, as these networks are highly sensitive to signal irregularities that can disrupt spike timing and pattern recognition, a challenge highlighted in research on EEG signal processing [64]. Moreover, the quality of the input signals—often affected by factors like electrode placement, patient movement, or hardware limitations—can significantly impact the reliability of SNN-based analyses, necessitating advanced preprocessing techniques [65]. Future directions should focus on developing standardised SNN frameworks, enhancing compatibility with existing systems, improving noise resilience through innovative filtering methods, and advancing signal quality assessment to unlock the full potential of SNNs in neuroimaging applications.

### 7.3. Methodological and Computational Challenges

The advancement of SNNs in neuroimaging hinges on progress across several critical fronts.

Developing improved training algorithms, such as surrogate gradient methods or biologically inspired learning rules, is paramount for achieving efficiency and effectiveness in complex neuroimaging tasks.

Incorporating biologically realistic models, including detailed neuron models and synaptic plasticity mechanisms, is essential for enhancing the accuracy and interpretability of SNNs, ultimately leading to a deeper understanding of brain function.

Creating multimodal integration techniques that effectively combine data from various neuroimaging modalities like fMRI, EEG, and MEG will facilitate a more comprehensive view of brain activity.

Enhancing the explainability of SNNs through research on Explainable AI (XAI) will make their decision-making processes more transparent and interpretable.

Exploring neuro-inspired computing paradigms, such as event-based processing or in-memory computing, can create more efficient and powerful SNNs for neuroimaging applications.

Developing clinical applications based on SNNs, including disease diagnosis, prognosis prediction, and treatment optimisation tools, can significantly improve patient care and outcomes.

Creating personalised SNN models that capture individual subjects’ unique brain activity patterns by tailoring the SNN architecture and parameters can improve the accuracy of diagnosis, prognosis, and treatment planning. Further, training SNNs is challenging due to non-differentiable spikes, which hinder gradient-based training; limited temporal memory that struggles with long-range dependencies; and sparse data representation, which can cause vanishing gradients and optimisation difficulties.

The development of neuromorphic hardware designed explicitly for SNNs will likely accelerate their adoption in neuroimaging, enabling the simulation of larger and more complex networks and pushing the boundaries of what is possible with this technology.

Developing emerging standards like Neural NIR could address NeuCube’s lack of standardised interfaces (Section 7.1), enabling seamless model sharing across SNN frameworks and neuromorphic hardware like Loihi2. Unlike ONNX, which is widely adopted for ANNs but requires adaptation for spike-based data, NIR natively supports SNN-specific features like spike trains and temporal dynamics, making it ideal for neuroimaging tasks such as EEG and fMRI analysis.

### 7.4. Ethical and Clinical Implications of Using SNN in Neuroimaging

Integrating SNNs in neuroimaging introduces ethical and clinical implications that warrant careful consideration. A primary concern is data privacy, as neuroimaging data, such as EEG or ECoG signals, often contains highly sensitive personal information about a patient’s neurological state. The use of SNNs, especially when implemented on neuromorphic hardware or cloud-based systems, raises the risk of unauthorised access or data breaches, necessitating robust encryption and compliance with regulations like GDPR or HIPAA, as emphasised in discussions on neuroscience data governance [66]. Another critical issue is algorithmic bias, which can arise if SNN models are trained on non-representative datasets, potentially leading to disparities in diagnostic accuracy across diverse populations such as varying age groups, ethnicities, or neurological conditions, a concern also raised in studies on AI fairness in healthcare [67]. This bias could exacerbate health inequities, particularly in underrepresented groups. Furthermore, the potential for misdiagnosis poses a significant clinical risk, as the complex, event-driven nature of SNNs may lead to errors in interpreting neural patterns, especially in noisy or low-quality datasets, potentially resulting in incorrect treatment recommendations, a risk noted in analyses of deep learning in neuroimaging [68]. Addressing these challenges requires the development of transparent SNN models with explainable outputs, rigorous validation against diverse datasets, and the establishment of ethical guidelines to ensure patient safety and equity in clinical applications of SNN-based neuroimaging.

### 7.5. Future Directions of Using SNN for Neuroimaging

Advancing spiking neural networks in neuroimaging requires focused efforts across several key areas: developing improved training algorithms for efficiency, incorporating biologically realistic models for accuracy and interpretability, creating multimodal integration methods for comprehensive brain activity understanding, enhancing explainability through XAI research, exploring neuro-inspired computing paradigms for efficiency gains, developing clinical applications for improved patient care, creating personalised SNN models for individual-specific analysis, and fostering the development of neuromorphic hardware to accelerate adoption and enable larger, more complex simulations.

Below, we highlight potential applications of NeuCube in neuroimaging and medicine, leveraging its unique reservoir-based architecture for predictive modelling and early diagnosis, personalised medicine, and real-time brain–computer interfaces.

#### 7.5.1. Predictive Modelling and Early Diagnosis

NeuCube offers a powerful tool for advancing neurological and mental health care through its ability to analyse brain activity patterns. It can predict disease progression in conditions like Alzheimer’s and Parkinson’s, enabling early interventions and personalised treatment. Furthermore, it can forecast treatment response based on individual brain characteristics, optimising treatment strategies. NeuCube also facilitates health risk assessment for conditions like stroke and dementia, allowing for proactive measures to promote healthy ageing. In neurological disorders, it aids in early diagnosis by detecting subtle changes in brain activity before clinical symptoms appear, leading to timely interventions.

Finally, NeuCube can assist in biomarker discovery by identifying specific brain activity patterns associated with disease onset or progression, which can be used for early diagnosis and disease monitoring.

#### 7.5.2. Personalised Medicine

It can track disease progression, predicting disease trajectory and complications to enable proactive interventions. By analysing longitudinal neuroimaging data, NeuCube optimises treatment by predicting individual responses and identifying optimal interventions. Furthermore, it allows for early diagnosis and prevention by detecting subtle changes in brain activity patterns that precede clinical symptoms. NeuCube also supports personalised health monitoring through continuous brain activity tracking, allowing for early detection of deviations and timely interventions. Finally, it contributes to understanding individual differences in brain activity and their relation to health outcomes, paving the way for more personalised healthcare approaches.

#### 7.5.3. Brain–Computer Interfaces

NeuCube offers significant potential across several domains by decoding and interpreting brain signals. It enables real-time control of prosthetics through translating movement intentions into control signals, fostering natural and intuitive control. Furthermore, it facilitates neurofeedback and rehabilitation by providing real-time feedback based on brain activity, aiding in training and recovery for neurological conditions. NeuCube can also power assistive technologies like communication devices and brain-controlled wheelchairs, enhancing interaction for individuals with disabilities. Human–computer interaction enables more intuitive control of computers and devices by decoding user intentions. Finally, it can revolutionise gaming and entertainment by allowing brain-controlled games and immersive experiences.

## 8. Conclusions

SNNs offer a viable approach to addressing some of the limitations of traditional ANNs in multimodal neuroimaging, particularly in processing spatiotemporal data, such as EEG and fMRI. For nontemporal data like MRI, our preliminary analysis of SNN simulators, including snnTorch and NeuCube, on datasets like ADNI demonstrates their unfitness to achieve comparable accuracy to ANNs for tasks like Alzheimer’s classification; therefore, an ANN-SNN hybrid approach to transfer learning is preferred, as shown in Section 6. NeuCube’s 3D brain-like architecture has proven effective in specific applications, such as EEG-based disease classification with 92% accuracy [20], and its energy-efficient design supports deployment on neuromorphic hardware like Loihi2, reducing power consumption by approximately 200× compared to GPU-based ANNs [36]. However, challenges remain, including the complexity of training SNNs due to non-differentiable spikes and the limited scalability of current neuromorphic platforms, which currently restrict their use to small-scale neuroimaging studies. For NeuCube, issues such as the lack of standardised interfaces and the need for large-scale validation on diverse patient cohorts hinder its widespread clinical adoption. Moving forward, targeted improvements, such as developing efficient training algorithms (e.g., surrogate gradients) and expanding hardware accessibility, could enhance SNN performance in neuroimaging. While NeuCube shows promise for personalised medicine and brain–computer interfaces, its full potential will depend on overcoming these technical and validation hurdles. It requires collaborative efforts between researchers and clinicians to integrate it into practical workflows.

## Figures and Tables

**Figure 1 bioengineering-12-00628-f001:**
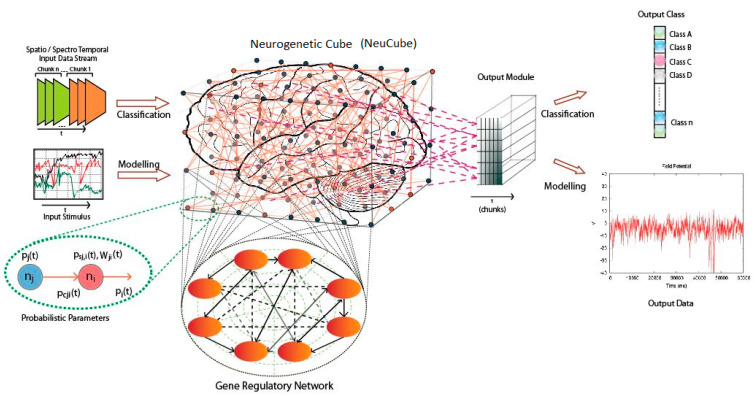
NeuCube rchitecture (from [18]).

**Figure 2 bioengineering-12-00628-f002:**
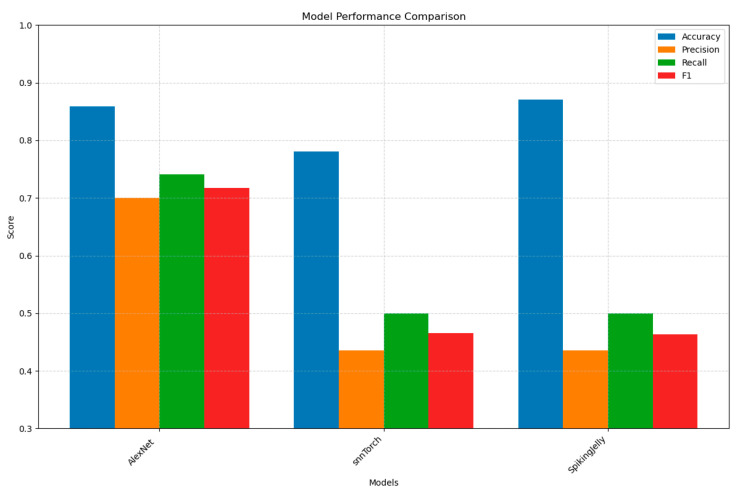
Performance metrics for training SNN simulators on vector-based data.

**Figure 3 bioengineering-12-00628-f003:**
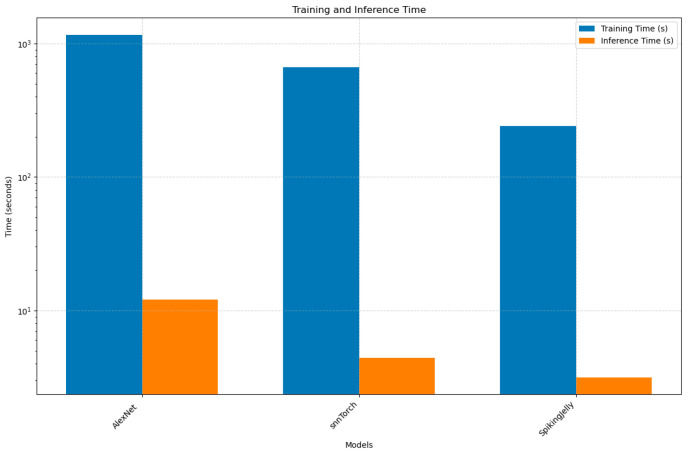
Training and inference time for SNN simulators on vector-based data.

**Table 1 bioengineering-12-00628-t001:** SNNs vs. ANNs.

FeatFure	SNNs	ANNs
Neuron Model	User biologically realistic spiking neurons (e.g., Leaky Integrate-and-Fire)	Simplified activation functions (e.g., ReLU, Sigmoid)
Information Encoding	Time-based spike trains (temporal coding)	Continuous values
Energy Efficiency	Highly efficient event-driven computation	Less efficient (always active)
Hardware Suitability	Optimized for neuromorphic chips (e.g., Loihi, SpiNNaker)	Runs on conventional GPUs, CPUs
Scalability	Scalable (e.g., Loihi, SpiNNaker)	Scalable but power-intensive
Training Method	Spike-Timing-Dependent Plasticity Surrogate gradients	Backpropagation (e.g., Stochastic Gradient Descent)
Temporal Dynamics	Captures precise timing information	Treats time as separate dimension
Accuracy	Lower on static data excels in temporal/event-based tasks	Higher on traditional (e.g., image, text) tasks
Applications	Brain-computer interfaces, edge AI, low-power robotics	Computer vision, NLP, general-purpose deep learning

**Table 2 bioengineering-12-00628-t002:** Feature comparison for SNN simulators.

Feature	snnTorch	SpikingJelly	Nengo DL	NeuCube	BindsNet
Neuron Model	LIF, Izhikevich	LIF, Adaptive LIF	LIF, H-H, Adaptive LIF	LIF	LIF, Izhikevich
Reservoir	No	Yes (custom)	Yes	Yes (3D grid)	No
Hierarchical (layers)	Yes (PyTorch)	Yes (modular)	Yes	Self-organised trajectories of neuronal clusters	Limited
Learning Algorithms	STDP, Surrogate Grad	STDP, BPTT	STDP, RL, PES	STDP, Evolving	STDP, BPTT
Neuromorphic	No	Loihi, Lynxi KA200	Loihi, SpiNNaker	Lohi, SpiNNaker	Loihi
Best For	Hybrid ANNs	Large-scale SNNs	Brain modeling	EEG/fMRI analysis	Prototyping

**Table 3 bioengineering-12-00628-t003:** Summary of SNN integration approaches.

Type	Method	Application	Ref.
SNN + deESN	SNN extracts spatio-temporal features CNN classifies or extracts spatial features Hybrid training (back propagation + STDP)	Neuroimaging analysis Brain Computer Interfaces	[46]
SNN + ESN	3D SNN (personalized), processes ECoG ESN classifies for motor control	Prosthetic control for paralysis	[58]
SNN + NeuroFuzzy	3D SNN extracts feature vectors, classified in a neuro-fuzzy model	Adaptive prosthetics	[59]

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
