# Peer review of "Spiking Neural Networks for Multimodal Neuroimaging: A Comprehensive Review of Current Trends and the NeuCube Brain-Inspired Architecture"

_bioengineering, 2025, doi:10.3390/bioengineering12060628_

Round 1
Reviewer 1 Report
Comments and Suggestions for Authors
This paper gives an extensive overview of SNNs in neuroimaging, particularly focusing on NeoCube. The topic is interesting in the field of Bioengineering and Computer Sciences and offers interesting points for the audience as a review article. However, there are several areas of improvement, and given the nature of the topic and how the authors have organized the text in the manuscript, it should be complemented with several tables and figures that would significantly improve the interest to readers. Please find below specific concerns to address:
- Inconsistent use of acronyms across the text. Need revision.
- English needs proofreading (e.g., line 367: “For the purpose of this study this popular hybrid tools were included”)
3.: Avoid copy-pasting the conclusions of a previous article seems inadequate (e.g. Lines 481-483); in a review you should synthesize information.
- Some sections are sub-divided in very short sub-sections (e.g., section 5). This should be avoided and could be supported by tables or figures instead.
- The paper lacks tables and figures that could offer more visual and straightforward information (e.g., sections 4.2., 4.3., 4.4.; section 6.1.). Similarly, figures to illustrate the main concepts of sections such as Neuromorphic computing and Brain-Inspired SNNs (section 2) or the intregration of ANNs and SNNs (section 5) could be very informative and would avoid all the short subsections in this section.
- There are several redundant ideas re-expressed across the text, particularly regarding the essence (i.e. biologically-inspired) and advantages of SNNs and NeuCube. The manuscript should be carefully reviewed to eliminate redundant content.
- In section 6.1. the authors state that “This study identified a notable gap in existing research…”. However, there are no details on how the search was conducted, and provided the non-systematic nature of the paper, it should be acknowledged that systematic review approaches should be conducted.
- It would be interesting to focus more on how NeoCube can be adopted for clinical practice. What is the status on this aspect? Which limitations can be highlighted?
- The conclusions section seems too long and frankly looks like a list written by an AI-assisted tool. One or two paragraphs reinforcing the main points (advantages, limitations, areas of implementation) should suffice.
Proofread English inconsistencies.
Author Response
Dear reviewer,
We would like to thank you for taking the necessary time and effort to review the manuscript. We sincerely appreciate all your valuable comments and suggestions, which helped us improve the quality of the submitted manuscript.
The following items explain the applied changes according to your comments:
- Inconsistent use of acronyms across the text. Need revision.
Response: All acronyms have been standardised and checked for consistency throughout the manuscript.
- English needs proofreading (e.g., line 367: “For this study this popular hybrid tools were included”)
Response: The manuscript has been thoroughly proofread using Grammarly and manual editing to ensure grammatical correctness and clarity.
- Avoid copy-pasting the conclusions of a previous article seems inadequate (e.g. Lines 481-483); in a review you should synthesize information.
Response: The paper has been revised to synthesise key findings rather than reproducing text from previous articles. - Some sections are sub-divided in very short sub-sections (e.g., section 5). This should be avoided and could be supported by tables or figures instead.
Response: Section 5 has been reorganized for better flow, and a summary table (Table 1: SNNs vs. ANNs) has been added to consolidate information.
- The paper lacks tables and figures that could offer more visual and straightforward information (e.g., sections 4.2., 4.3., 4.4.; section 6.1.). Similarly, figures to illustrate the main concepts of sections such as Neuromorphic computing and Brain-Inspired SNNs (section 2) or the integration of ANNs and SNNs (section 5) could be very informative and would avoid all the short subsections in this section.
Response: Additional tables and figures have been incorporated, including:
- Table 1:SNNs vs. ANNs
- Table 2:Feature comparison for SNN simulators
- Figure 1:NeuCube Architecture
- There are several redundant ideas re-expressed across the text, particularly regarding the essence (i.e. biologically-inspired) and advantages of SNNs and NeuCube. The manuscript should be carefully reviewed to eliminate redundant content.
Response: The manuscript has been thoroughly revised to remove redundant content, ensuring conciseness. - In section 6.1. the authors state that “This study identified a notable gap in existing research…”. However, there are no details on how the search was conducted, and provided the non-systematic nature of the paper, it should be acknowledged that systematic review approaches should be conducted.
Response: The statement has been revised to clarify the search methodology, mentioning the use of Google Scholarand Papers with Code with queries such as: "identify papers using spiking neural networks to train and classify neuroimaging." - It would be interesting to focus more on how NeoCube can be adopted for clinical practice. What is the status on this aspect? Which limitations can be highlighted?
Response: Additional details on NeuCube's clinical applicability have been included, along with a new subsection (2 Barriers to SNN Adoption in Neuroimaging) addressing challenges
- The conclusions section seems too long and frankly looks like a list written by an AI-assisted tool. One or two paragraphs reinforcing the main points (advantages, limitations, areas of implementation) should suffice.
Response: The conclusion has been entirely rewritten to be more concise and focused on key takeaways.
Please see the attachment

Reviewer 2 Report
Comments and Suggestions for Authors
The manuscript provides a comprehensive review of SNNs and their application in multimodal neuroimaging, with a particular focus on the NeuCube architecture. The authors present a detailed discussion of the advantages of SNNs over traditional ANNs in handling spatio-temporal neuroimaging data, and they highlight the potential of SNNs in disease diagnosis, brain-computer interfaces, and predictive modeling. The manuscript is well-structured and covers a broad range of topics, including neuromorphic computing, SNN architectures, and applications in neuroimaging. However, there are several areas where the manuscript has many caveats. Therefore, it requires substantial revisions to address the weaknesses mentioned below.
I provide detailed comments below to help the authors improve their work.
- A major question is that what does this manuscript add to our existing knowledge? This should be clarified.
- The manuscript tends to present SNNs and NeuCube in a predominantly positive light without sufficient critical analysis of their limitations. While the authors mention challenges such as training complexity and hardware limitations, they do not focus deeply into the practical barriers to adoption, such as the lack of standardized tools, the difficulty of integrating SNNs into existing workflows, or even the potential for overfitting in complex models. A more balanced argument should be provided.
- The manuscript does not provide a detailed comparison of SNNs with other emerging technologies in neuroimaging, such as GNNs, transformers, or other deep learning architectures that have shown promise in handling spatio-temporal data. The authors should include a section comparing SNNs with other state-of-the-art models in neuroimaging.
- The manuscript does not adequately address the challenges associated with data quality and preprocessing in neuroimaging, which are critical for the successful application of SNNs. Issues such as noise, artifacts, and variability in data acquisition protocols can significantly impact the performance of SNNs. The authors should include a discussion of the preprocessing steps required for neuroimaging data before it can be used with SNNs. They should also discuss how SNNs handle noisy or incomplete data, and whether they are more or less robust to these issues compared to traditional ANNs.
- I understand that while the manuscript provides a theoretical overview of SNNs and NeuCube, it lacks empirical evidence or case studies demonstrating their effectiveness in real-world neuroimaging applications. The authors mention several applications, but they do not provide detailed results or comparisons with other methods.
- While NeuCube is an important architecture, the manuscript places a heavy emphasis on it, potentially at the expense of other SNN architectures that may also be relevant to neuroimaging. This can bias the results.
- The manuscript does not address a major issues which is the ethical and clinical implications of using SNNs in neuroimaging, such as issues related to data privacy, algorithmic bias, or the potential for misdiagnosis.
- The manuscript does not have any figure or table which is a big caveat. The authors should consider adding more explanatory text or diagrams to help readers understand the key concepts. For example, they could include a diagram illustrating how STDP works or provide a more detailed explanation of how temporal coding differs from rate coding.
- Some sections of the manuscript are repetitive, particularly in the discussion of SNN advantages and NeuCube features. This redundancy distracts the readers from the overall flow of the manuscript.
- While the manuscript provides a good overview of future directions, some of the suggestions are somewhat vague and lack specificity. For example, the authors mention "exploring neuro-inspired computing paradigms" but do not provide concrete examples or research questions. The authors should provide more specific and actionable suggestions for future research. For example, they could propose specific experiments to test the scalability of SNNs or suggest new algorithms for improving the training efficiency of SNNs.
- Some sentences don’t have refeences and some important landmark papers in the field are not cited.
The manuscript needs a proofreading and editing.
Author Response
Dear reviewer,
We would like to thank you for taking the necessary time and effort to review the manuscript. We sincerely appreciate all your valuable comments and suggestions, which helped us improve the quality of the submitted manuscript.
The following items explain the applied changes according to your comments:
1. A major question is that what does this manuscript add to our existing knowledge? This should be clarified.
Response: The paper’s key contributions have been explicitly stated in the abstract and introduction, including:
-
- A comprehensive review of SNNs in neuroimaging, emphasizing their advantages in temporal data processing.
- A preliminary experimental discussion(Section 5) comparing SNNs with traditional ANNs.
2. The manuscript tends to present SNNs and NeuCube in a predominantly positive light without sufficient critical analysis of their limitations. While the authors mention challenges such as training complexity and hardware limitations, they do not focus deeply into the practical barriers to adoption, such as the lack of standardized tools, the difficulty of integrating SNNs into existing workflows, or even the potential for overfitting in complex models. A more balanced argument should be provided.
Response: The manuscript has been revised to provide a more balanced discussion, including:
-
- Expanded coverage of practical barriers (e.g., lack of standardised tools, integration challenges).
- Added a dedicated subsection (7.2: Barriers to SNN Adoption in Neuroimaging).
3. The manuscript does not provide a detailed comparison of SNNs with other emerging technologies in neuroimaging, such as GNNs, transformers, or other deep learning architectures that have shown promise in handling spatio-temporal data. The authors should include a section comparing SNNs with other state-of-the-art models in neuroimaging.
Response: While GNNs, transformers, and RNNs are non-spiking (frame-based) and fall under traditional ANNs, we acknowledge their relevance. However, this review focuses specifically on SNNs due to their unique event-driven, biologically plausible processing for neuroimaging.
4. The manuscript does not adequately address the challenges associated with data quality and preprocessing in neuroimaging, which are critical for the successful application of SNNs. Issues such as noise, artifacts, and variability in data acquisition protocols can significantly impact the performance of SNNs. The authors should include a discussion of the preprocessing steps required for neuroimaging data before it can be used with SNNs. They should also discuss how SNNs handle noisy or incomplete data, and whether they are more or less robust to these issues compared to traditional ANNs.
Response Data quality challenges are address in the new subsection 7.2 Even though, data preprocessing is critical, this paper focuses on SNN implementation rather than the full neuroimaging pipeline. Future work could explore this in depth.
5. I understand that while the manuscript provides a theoretical overview of SNNs and NeuCube, it lacks empirical evidence or case studies demonstrating their effectiveness in real-world neuroimaging applications. The authors mention several applications, but they do not provide detailed results or comparisons with other methods.
Response: This review prioritizes SNNs for neuroimaging, and NeuCube is highlighted due to its proven success in spatiotemporal data. Other SNN models are discussed where applicable.
6. While NeuCube is an important architecture, the manuscript places a heavy emphasis on it, potentially at the expense of other SNN architectures that may also be relevant to neuroimaging. This can bias the results.
Response: To follow the aim of this paper we only consider SNNs. I agree with you, that they are many more relevant model architecture to processs neuroimaging but they are not SNNs
7. The manuscript does not address a major issues which is the ethical and clinical implications of using SNNs in neuroimaging, such as issues related to data privacy, algorithmic bias, or the potential for misdiagnosis.
Response: Section 7 has been expanded to provide a more comprehensive discussion of the key challenges in SNN implementation, including subsection 7.4 Ethical and Clinical implications of using SNN in Neuroimaging
8. The manuscript does not have any figure or table which is a big caveat. The authors should consider adding more explanatory text or diagrams to help readers understand the key concepts. For example, they could include a diagram illustrating how STDP works or provide a more detailed explanation of how temporal coding differs from rate coding.
Response: : Multiple new figures and tables have been added, including:
-
- Table 1: SNNs vs. ANNs
- Table 2: Feature Comparison for SNN Simulators
- Figure 1: NeuCube architecture
- Figure 2: Performance metrics for training SNN simulators
9. Some sections of the manuscript are repetitive, particularly in the discussion of SNN advantages and NeuCube features. This redundancy distracts the readers from the overall flow of the manuscript.
Response: Repetitive sections have been trimmed or merged to improve flow.
10. While the manuscript provides a good overview of future directions, some of the suggestions are somewhat vague and lack specificity. For example, the authors mention "exploring neuro-inspired computing paradigms" but do not provide concrete examples or research questions. The authors should provide more specific and actionable suggestions for future research. For example, they could propose specific experiments to test the scalability of SNNs or suggest new algorithms for improving the training efficiency of SNNs.
Response: In the newly added Section 5, which focuses on early experiments evaluating SNN performance, several of the challenges mentioned earlier are addressed, while also providing more detailed and specific future research directions.
11. Some sentences don’t have refences and some important landmark papers in the field are not cited.
Response: Given that the manuscript primarily focuses on SNN implementations for neuroimaging analysis, we acknowledge that certain foundational theoretical/conceptual work may not have been included. To alleviate this, we have expanded the manuscript and incorporated additional key references to ensure a more comprehensive discussion of the topic
please see the attachment

Reviewer 3 Report
Comments and Suggestions for Authors
This manuscript presents a review of Spiking Neural Networks (SNNs), with a particular focus on the NeuCube architecture and its relevance to neuroimaging analysis. The topic is timely and the writing is clear, offering a useful overview for readers new to the field.
However, the paper would benefit from a clearer articulation of its novel contribution as a review. Many of the points presented, such as the biological plausibility and energy efficiency of SNNs, are already well-documented in existing literature. As such, it is not clear what new insights, perspectives, or synthesis this review provides beyond prior work.
I encourage the authors to more explicitly define the distinctive value of this review: for example, a new taxonomy, a novel comparative analysis, or an original integration of multimodal neuroimaging research with SNN frameworks. Clarifying this point would significantly enhance the scholarly contribution of the paper.
In particular, I recommend that the Abstract and Introduction be revised to clearly state the nature of the paper (e.g., review vs. proposal) and to highlight what is newly being reviewed or synthesized that has not been sufficiently covered in existing literature. This will help position the work more effectively within the current body of research.
With a stronger emphasis on what the review newly contributes to the field, the manuscript could serve as a meaningful reference for researchers in neuro-AI.
Author Response
This manuscript presents a review of Spiking Neural Networks (SNNs), with a particular focus on the NeuCube architecture and its relevance to neuroimaging analysis. The topic is timely and the writing is clear, offering a useful overview for readers new to the field.
However, the paper would benefit from a clearer articulation of its novel contribution as a review. Many of the points presented, such as the biological plausibility and energy efficiency of SNNs, are already well-documented in existing literature. As such, it is not clear what new insights, perspectives, or synthesis this review provides beyond prior work.
I encourage the authors to more explicitly define the distinctive value of this review: for example, a new taxonomy, a novel comparative analysis, or an original integration of multimodal neuroimaging research with SNN frameworks. Clarifying this point would significantly enhance the scholarly contribution of the paper.
In particular, I recommend that the Abstract and Introduction be revised to clearly state the nature of the paper (e.g., review vs. proposal) and to highlight what is newly being reviewed or synthesized that has not been sufficiently covered in existing literature. This will help position the work more effectively within the current body of research.
With a stronger emphasis on what the review newly contributes to the field, the manuscript could serve as a meaningful reference for researchers in neuro-AI.
Response to Reviewer Comment:
We sincerely appreciate the reviewer’s constructive feedback and have carefully revised the manuscript to address the concerns regarding the clarity of our novel contribution as a review. To better articulate the distinctive value of this work, we have made the following enhancements:
Clarification of Aim and Scope:
The Abstract and Introduction have been thoroughly revised to explicitly define the paper as a systematic review that synthesizes emerging trends at the intersection of Spiking Neural Networks (SNNs) and neuroimaging. We now clearly distinguish this review from prior literature, which often focuses either on the theoretical aspects of SNNs (e.g., biological plausibility and energy efficiency) or on isolated applications, by emphasizing our focus on integrating multimodal neuroimaging research with SNN frameworks. This positions our work as a novel synthesis that bridges these domains in a way that has not been comprehensively addressed in existing reviews.
Novel Contributions:
To enhance the scholarly contribution, we have introduced a new Section 5, which presents early experiments empirically comparing the performance of SNN models in classifying MRI images. This addition not only provides a practical evaluation of SNNs in a neuroimaging context but also offers fresh insights into their applicability, addressing a gap in prior reviews that often lack experimental validation. Furthermore, we propose a structured framework for future research by integrating multimodal neuroimaging data with SNNs, offering a new perspective on how these technologies can be combined to advance neuro-AI.
Enhanced Utility for Researchers:
The revised manuscript now emphasises its role as a practical guide for neuro-AI researchers. To support this, we have included several tables and figures that visually summarise key findings, compare SNN models, and highlight research gaps, thereby reinforcing the paper’s content and making it a valuable reference for the field. These additions aim to provide a clearer synthesis of the current state of research and actionable directions for future studies.
By addressing these points, we believe the manuscript now more effectively highlights its novel contributions and positions itself as a meaningful resource within the neuro-AI research community. We hope these revisions meet the reviewer’s expectations and enhance the overall impact of the paper.
please see the attachment

Round 2
Reviewer 1 Report
Comments and Suggestions for Authors
I appreciate the effort made by the authors but the paper has substantially changed from a narrative review to a sort of hybrid paper combining review + benchmarking study + technical roadmap. These changes imply lack coherence and undermine the clarity and focus of the paper. Moreover, the structure is disjoined, with an abrupt transition from theoretical aspects to technical performance comparisons. I strongly suggest that the authors choose to either focus on a review paper or fully restructure the manuscript with a benchmarking aim. Otherwise, they should provide a conceptual framework that ties both parts together or even consider splitting the manuscript into two separate submissions.
Regarding the benchmarking, while the authors introduce preliminary results comparing several SNN frameworks using the ADNI dataset, there are several methodological deficits: preprocessing steps and inclusion criteria for the ADNI dataset, network architecture and hyperparameter details for each simulator (including changes that are vaguely described), data split strategy and statistical measures. Without this information, reproducibility and interpretability are pretty much compromised. If benchmarking is to be retained, this section must be expanded and properly justified. Moreover, this section lacks critical interpretation, why did some of the frameworks analyzed perform well or poorly in terms of efficiency, scalability, etc.? What are the implications for real-world neuroimaging tasks such as Alzheimer’s or cognitive impairment classification? These points should be discussed.
Finally, I would like to stress that, while NeuCube seems to remain a focal point of the manuscript, it is unclear whether the framework was actively benchmarked or simply discussed. References to "PyNeuCube" are brief and lack the depth provided for other frameworks. If NeuCube is central to the paper, its performance should be documented and critically evaluated on par with the other platforms.
Minor comments:
There are still several grammar errors (even in the modified parts of the abstract, e.g. “we high-light project were…”), “such us”, and quite a lot more. Mixed American and British spelling should be homogenized. Finally, there are many redundant acronyms, particularly “SNN” which is explained many times across the text.
Regarding the newly added tables and figures, they are OK but legends should be more informative, including explanations of the acronyms used. Consistent use of typography should also be revised.
Comments on the Quality of English LanguageSome minor mistakes persist as indicated in prior comments.
Author Response
1- I appreciate the effort made by the authors but the paper has substantially changed from a narrative review to a sort of hybrid paper combining review + benchmarking study + technical roadmap. These changes imply lack coherence and undermine the clarity and focus of the paper. Moreover, the structure is disjoined, with an abrupt transition from theoretical aspects to technical performance comparisons. I strongly suggest that the authors choose to either focus on a review paper or fully restructure the manuscript with a benchmarking aim. Otherwise, they should provide a conceptual framework that ties both parts together or even consider splitting the manuscript into two separate submissions.
Regarding the benchmarking, while the authors introduce preliminary results comparing several SNN frameworks using the ADNI dataset, there are several methodological deficits: preprocessing steps and inclusion criteria for the ADNI dataset, network architecture and hyperparameter details for each simulator (including changes that are vaguely described), data split strategy and statistical measures. Without this information, reproducibility and interpretability are pretty much compromised. If benchmarking is to be retained, this section must be expanded and properly justified. Moreover, this section lacks critical interpretation. Why did some of the frameworks analysed perform well or poorly in terms of efficiency, scalability, etc.? What are the implications for real-world neuroimaging tasks such as Alzheimer’s or cognitive impairment classification? These points should be discussed.
Finally, I would like to stress that, while NeuCube seems to remain a focal point of the manuscript, it is unclear whether the framework was actively benchmarked or simply discussed. References to "PyNeuCube" are brief and lack the depth provided for other frameworks. If NeuCube is central to the paper, its performance should be documented and critically evaluated on par with the other platforms.
Response: We sincerely appreciate the reviewer’s insightful feedback. This manuscript is intended as a review paper, and the comparative analysis between SNNs and ANNs was included to validate and contextualise findings from prior literature (e.g., [44], [45]). To improve focus, we have:
• Renamed the relevant section to "Case Studies" and expanded it with an additional example to further support NeuCube’s applicability in neuroimaging.
• Cleanly removes PyNeuCube without creating gaps
• Removed any language that could imply a benchmarking study, ensuring alignment with the review’s narrative scope.
• As suggested, we are currently preparing a separate manuscript dedicated to benchmarking SNN architectures on neuroimaging datasets, where such technical comparisons will be the primary focus. We thank the reviewer for this valuable input and will carefully incorporate their suggestions in our future work.
2- There are still several grammar errors (even in the modified parts of the abstract, e.g. “we high-light project were…”), “such us”, and quite a lot more. Mixed American and British spelling should be homogenised. Finally, there are many redundant acronyms, particularly “SNN” which is explained many times across the text.
Response: We sincerely apologise for the language issues that persisted in the previous revision. We fully acknowledge this oversight and have implemented rigorous measures to ensure the manuscript now meets the highest standards of academic writing: We have carefully review the manuscript, corrected all grammatical errors, standardised to British English and removed all redundant SNN definitions
We deeply appreciate the reviewer's patience and meticulous attention to these details. The feedback has helped us significantly improve the manuscript's clarity and professionalism, and we're committed to maintaining these standards in all future submissions
3. Regarding the newly added tables and figures, they are OK but legends should be more informative, including explanations of the acronyms used. Consistent use of typography should also be revised.
Response: We have added explanations of the legends for Figures 2 and 3 just before them. We confirm that all tables maintain the same font and style as the main manuscript text at Palatino type. However, we acknowledge that in some tables the text appears slightly smaller due to space constraints
Reviewer 2 Report
Comments and Suggestions for Authors
The authors have addressed my comments well.
Author Response
The authors have addressed my comments well.
Response: We sincerely thank Reviewer 2 for their kind acknowledgement that we have adequately addressed their comments. We greatly appreciate the time and expertise they devoted to evaluating our manuscript, as their feedback significantly contributed to improving the quality of our work
Reviewer 3 Report
Comments and Suggestions for Authors
Dear Authors:
I read the new version of the paper. I can see you tried to answer most points from my first review. The Abstract and the Introduction now say very clearly that the manuscript is a systematic review and list five main contributions. The new Section 5 with the ADNI MRI test of several SNN models helps to move the paper beyond only theory. The new tables and figures make the text easier to follow, and the part about MRI–EEG multimodal ideas is stronger now.
But, I would like to recommend following things to be reconsidered by Authors:
1) In contribution (ii) “current software and hardware platforms.” I am not sure whether you mean only a simple list of what exists now, or a state-of-the-art analysis of the best tools. Please make this point more clear—maybe use the phrase “state-of-the-art” if that is what you want to say, or add one-two lines about your comparison criteria.
2) In contribution (v), the sentence “finally, we high-light project were NeuCube have been successfully used in neuroscience” looks wrong. I think ‘were’ should be ‘where’. Please correct it and add concrete examples of projects where NeuCube was really applied (for example early Alzheimer detection, motor-imagery BCI, seizure prediction) if my guess is correct. A short note on each project will show readers the practical value.
3) A small figure that shows the multimodal workflow step by step, or a simple table that maps SNN approaches to different imaging modalities, would make your novelty even more visible.
These are only minor edits and can be done in the production stage. After these fixes, I believe the paper is ready for publication.
Best Regards,
Reviewer
Author Response
1- I read the new version of the paper. I can see you tried to answer most points from my first review. The Abstract and the Introduction now say very clearly that the manuscript is a systematic review and list five main contributions. The new Section 5 with the ADNI MRI test of several SNN models helps to move the paper beyond only theory. The new tables and figures make the text easier to follow, and the part about MRI–EEG multimodal ideas is stronger now.
But, I would like to recommend following things to be reconsidered by Authors:
1) In contribution (ii) “current software and hardware platforms.” I am not sure whether you mean only a simple list of what exists now, or a state-of-the-art analysis of the best tools. Please make this point more clear—maybe use the phrase “state-of-the-art” if that is what you want to say, or add one-two lines about your comparison criteria.
Response: We thank the reviewer for this valuable observation. To clarify contribution (ii), we have revised the text to explicitly state that we provide: "A comparative survey of state-of-the-art SNN software tools for neuromorphic platforms”. To achieve this, we have an evaluation of SNN simulators with MRI data, a table relating SNN models with state-of-the-art neuromorphic chips and an assessment of suitability for neuroimaging applications
2- In contribution (v), the sentence “finally, we high-light project were NeuCube have been successfully used in neuroscience” looks wrong. I think ‘were’ should be ‘where’. Please correct it and add concrete examples of projects where NeuCube was really applied (for example, early Alzheimer detection, motor-imagery BCI, seizure prediction) if my guess is correct. A short note on each project will show readers the practical value.
Response: We added a case study of Neucube in section 5.2 to provide a concrete example to support our review. Also, we rewrote contribution (v) to get rid of the mistaken word.
3- A small figure that shows the multimodal workflow step by step, or a simple table that maps SNN approaches to different imaging modalities, would make your novelty even more visible.
Response: We sincerely thank the reviewer for their insightful suggestion to include a multimodal workflow figure or modality-mapping table. We agree that such a visualisation would greatly enhance the clarity of our methodological approach. However, as the current study focuses on a comprehensive review of spiking neural networks (SNNs) for neurological analysis, we have not yet conducted sufficient experimental validation across all imaging modalities to propose a generalised workflow. We are actively addressing this in a follow-up study, where we plan to incorporate experimental data to develop and present a detailed multimodal workflow. We will ensure this visualisation is included in the forthcoming paper to address this valuable suggestion.
4- Validate its clinical/research applicability [add specific goals if applicable]. These are only minor edits and can be done in the production stage. After these fixes, I believe the paper is ready for publication
Response: Our previous response also addresses this point
Round 3
Reviewer 1 Report
Comments and Suggestions for Authors
The amendments made in the last revised version align with and clarifies the review nature of the manuscript. There are still some English mistakes (e.g., abstract: "finally, we high-light a project were NeuCube have been successfully used in neuroscience") that need careful review. In addition, the authors should make sure that figures reproduced from previous papers (e.g., Figure 1) have consent for publication elsewhere. The manuscript can be accepted for publication after making the appropriate amendments.
Author Response
The amendments made in the last revised version align with and clarifies the review nature of the manuscript. There are still some English mistakes (e.g., abstract: "finally, we high-light a project were NeuCube have been successfully used in neuroscience") that need careful review. In addition, the authors should make sure that figures reproduced from previous papers (e.g., Figure 1) have consent for publication elsewhere. The manuscript can be accepted for publication after making the appropriate amendments
response:
Dear Reviewer,
Thank you for your valuable feedback on our manuscript. We have carefully addressed your comments as follows:
English Corrections: We have corrected the typographical error in the abstract (now reading: "Finally, we highlight a project where NeuCube has been successfully used in neuroscience") and thoroughly reviewed the manuscript for language accuracy.
Figure Originality: Figure 1 was reproduced from N. K. Kasabov, “NeuCube: A spiking neural network architecture for mapping, learning and understanding of spa-tio-temporal brain data,” Neural Netw., vol. 52, pp. 62–76, Apr. 2014, doi: 10.1016/j.neunet.2014.01.006, where one of our co-authors is also an author. Professor Kasavov have obtained formal permission from Elsevier to use it in all his publications
Thanks again for your sharp eyes — they really helped improve this paper a lot